# Older people's experience of the partial lockdown during the COVID-19 pandemic in Switzerland: a cross-sectional study

Daphné Märki-Germann ![ORCID],[1] Laurence Seematter-Bagnoud,[1,2] Sarah Fustinoni,[2] Julia Spaltenstein,[1,2] Christophe Bula,[1] Yves Henchoz[2]

[1]Department of Geriatrics and Geriatric Rehabilitation, Lausanne University Hospital, Lausanne, Switzerland
[2]Department of Epidemiology and Health Systems, Centre for Primary Care and Public Health (Unisanté), University of Lausanne, Lausanne, Switzerland

**Correspondence to**
Daphné Märki-Germann;
daphne.marki-germann@unil.ch

## ABSTRACT

**Objectives** This study aimed to evaluate older people's experience of a COVID-19 partial lockdown (16 March–11 May 2020) in Lausanne, Switzerland.

**Setting and participants** Community-dwelling participants of the Lausanne cohort (Lc65+) in 2020, aged 71–86 years (n=2642).

**Design and outcome** This cross-sectional study was nested within the Lc65+ longitudinal study. A specific COVID-19 questionnaire was sent on 17 April 2020 to evaluate participants' experience of the lockdown (outcome). Multinomial logistic regression models were used to determine the sociodemographic, living environment, health and social factors associated.

**Results** Out of 2642 participants, 67.8% described the lockdown as 'somewhat' difficult (reference group), 21.5% as 'not at all' difficult (positive) and 10.7% as 'very or extremely' difficult (negative). The relative risk of a positive experience was higher in participants living alone (relative risk ratio, RRR=1.93, 95% CI 1.52 to 2.46) or in a house (RRR=1.49, 1.03 to 2.16); lower in those who reported fear of falling (RRR=0.68, 0.54 to 0.86), functional difficulties (RRR=0.78, 0.61 to 0.99), feeling of loneliness (RRR=0.67, 0.49 to 0.91), unfamiliarity with communication technologies (RRR=0.69, 0.52 to 0.91), usual social support (RRR=0.71, 0.50 to 0.93), previous participation in group activities (RRR=0.74, 0.59 to 0.92) and among women (RRR=0.75, 0.59 to 0.95). The relative risk of a negative experience was higher in participants with fear of falling (RRR=1.52, 1.07 to 2.15), and lower in those who had a terrace/garden (RRR=0.66, 0.44 to 0.99) and owned a dog (RRR=0.32, 0.11 to 0.90).

**Conclusions** Only one in 10 participants experienced the lockdown as very or extremely difficult. Specific interventions targeting vulnerability factors, such as fear of falling, could lessen the impact of any future similar situation.

## STRENGTHS AND LIMITATIONS OF THIS STUDY

⇒ This study focused on the experience of the COVID-19 partial lockdown in a large sample of community-dwelling older people.
⇒ The design of the study allowed linking data collected during the lockdown with key characteristics routinely assessed in the Lausanne cohort longitudinal study, and to highlight the ones that were independently associated with a positive and a negative experience of the lockdown.
⇒ Responses were obtained from postal—rather than online—questionnaires, allowing to include individuals who are uncomfortable with the use of communication technology.
⇒ The most vulnerable participants (7.5% of the total sample) had to be excluded from the study due to missing data or institutionalisation.
⇒ Most data stem from the 2019 questionnaire, and we cannot formally exclude minor changes in some participants' situation in 2020.

people because of their increased vulnerability to severe COVID-19 infection and hospitalisation.

Concerns rapidly emerged regarding the psychological impact of the lockdown. Early studies conducted during the first outbreak in China and the USA showed increased anxiety and depressive symptoms in the context of stay-at-home orders.[1–3] Overall, nearly half the general population reported moderate to severe levels of distress during the first epidemic outbreak.[3–6] These early studies suggested that people aged 60 years and over were particularly at risk of such adverse psychological impact.[3]

The partial lockdown deprived socially active older people of their usual contact with society. Older people who already were less active because of health problems also faced the fear of being infected through contacts with in-home care staff.[7]

## INTRODUCTION

In Switzerland, the Federal Council ordered a partial lockdown period lasting from 16 March to 11 May 2020 to prevent the spread of the highly contagious SARS-CoV-2 virus. Strong recommendations to stay strictly at home were specifically directed towards older

Several later studies conducted in other countries reported information about the early impact of the COVID-19 pandemic on anxiety and depressive symptoms in the general population.[8–10] Results showed a significantly increasing prevalence of mental distress during the first months of the epidemic, specifically of anxiodepressive and post-traumatic stress disorder symptoms.

Other studies investigated the sociodemographic and health predictors of depression and anxiety symptoms associated with the epidemic and the stay-at-home orders in the general population.[10 11] However, few specifically focused on the older population and none on the specific conditions of a partial lockdown. For instance, it remains unclear whether age itself is a risk factor for a negative experience or, reversely, whether it could be a protective factor due to resilience and accommodative strategies.[12 13] Similarly, the potential mediating effect of mobility and functional capacities has not been investigated.

A study conducted in Zürich, Switzerland, on 99 people aged 65 years and over showed that their level of well-being decreased as their feeling of loneliness increased during the first 4 weeks of the partial lockdown.[14] Another Swiss study conducted in the Ticino Canton on 19 community-dwelling older people revealed the latter's ambivalent feelings about the lockdown; some of them expressed their feeling of exclusion because older people were categorised as vulnerable.[15]

This study primarily aimed to evaluate how a large sample of older people experienced the partial lockdown during the first wave of the SARS-CoV-2 epidemic in Switzerland. Since social distancing measures particularly targeted the older population, but the lockdown was less strict in Switzerland than in many other countries, our hypothesis was that most participants would report a neutral experience, judging this period as somewhat difficult, whereas two smaller subgroups would report more extreme feelings and consider their experience as either negative (judging the period as very difficult) or positive (judging the period as not difficult at all).

A second aim was to study participants' characteristics associated with these more extreme feelings about their experience of the partial lockdown. More specifically, this study sought to investigate the association of four categories of participants' characteristics (sociodemographic, living environment, health status and social status) with a positive, respectively negative, experience of the lockdown.

We hypothesised an association between vulnerability factors (older age, lower education, higher comorbidities, as well as functional, cognitive and affective impairments) and a higher risk of a negative experience. By contrast, we hypothesised that participants who lived in a more comfortable environment (in a house, with access to a terrace/garden) would be more likely to report a positive experience. Finally, our hypotheses were mixed regarding social factors, as we expected that participants having less usual social support would be more likely to

report a negative experience, whereas those owning pet animals might report a positive experience.

## METHODS
### Study design and participants
This study is a cross-sectional analysis nested within the Lausanne cohort 65+ (Lc65+) longitudinal study,[16 17] initially designed to investigate the development of age-related frailty. Three samples of non-institutionalised individuals living in Lausanne (Switzerland) were successively enrolled in Lc65+ at the same age of 65–70 years in 2004 (birth years 1934–1938), 2009 (1939–1943) and 2014 (1944–1948).

For this cross-sectional analysis, data on participants' characteristics stem from the Lc65+ recruitment (2004, 2009 and 2014, respectively) and yearly follow-up (2016, 2019 and 2020) questionnaires. Specific details regarding the process of Lc65+ participants enrolment are available on the original study articles.[16 17]

A separate COVID-19 specific questionnaire was sent by postal mail to 3087 Lc65+ eligible participants on 17 April 2020 (details available on figure 1 for participants' selection). The questionnaires relating to their experience of the partial lockdown were completed during the lockdown to exclude memory bias. As a simplification measure, the 'partial lockdown' will henceforth be referred to as 'lockdown'.

The Lc65+ study protocol, including the COVID-19 specific questionnaire, was approved by the ethical committee for human's research in the canton of Vaud (Protocol No. 19/04). Written informed consent was obtained from the participants.

### Measurement of self-reported experience of the lockdown (outcome)
The study's outcome derived from participants' answer to the following question in the COVID-19 specific questionnaire: 'To what extent is the lockdown currently difficult to live with?'

According to their answers, the participants' experience of the lockdown was defined as (a) *Positive* in participants who answered 'not difficult at all', (b) *Neutral* (reference group) in participants who answered 'somewhat difficult' and (c) *Negative* in those who answered 'very difficult' or 'extremely difficult'. Thus, the term 'positive' refers to participants who reported a more positive experience of the lockdown than other participants and does not imply that the lockdown was a positive experience.

### Measurement of participants' characteristics
Variables with hypothesised associations with participants' experience of the lockdown (outcome) were identified in the four following domains:

### Sociodemographic
Sociodemographic variables included age (at the end of year 2020), gender, education level (low: compulsory

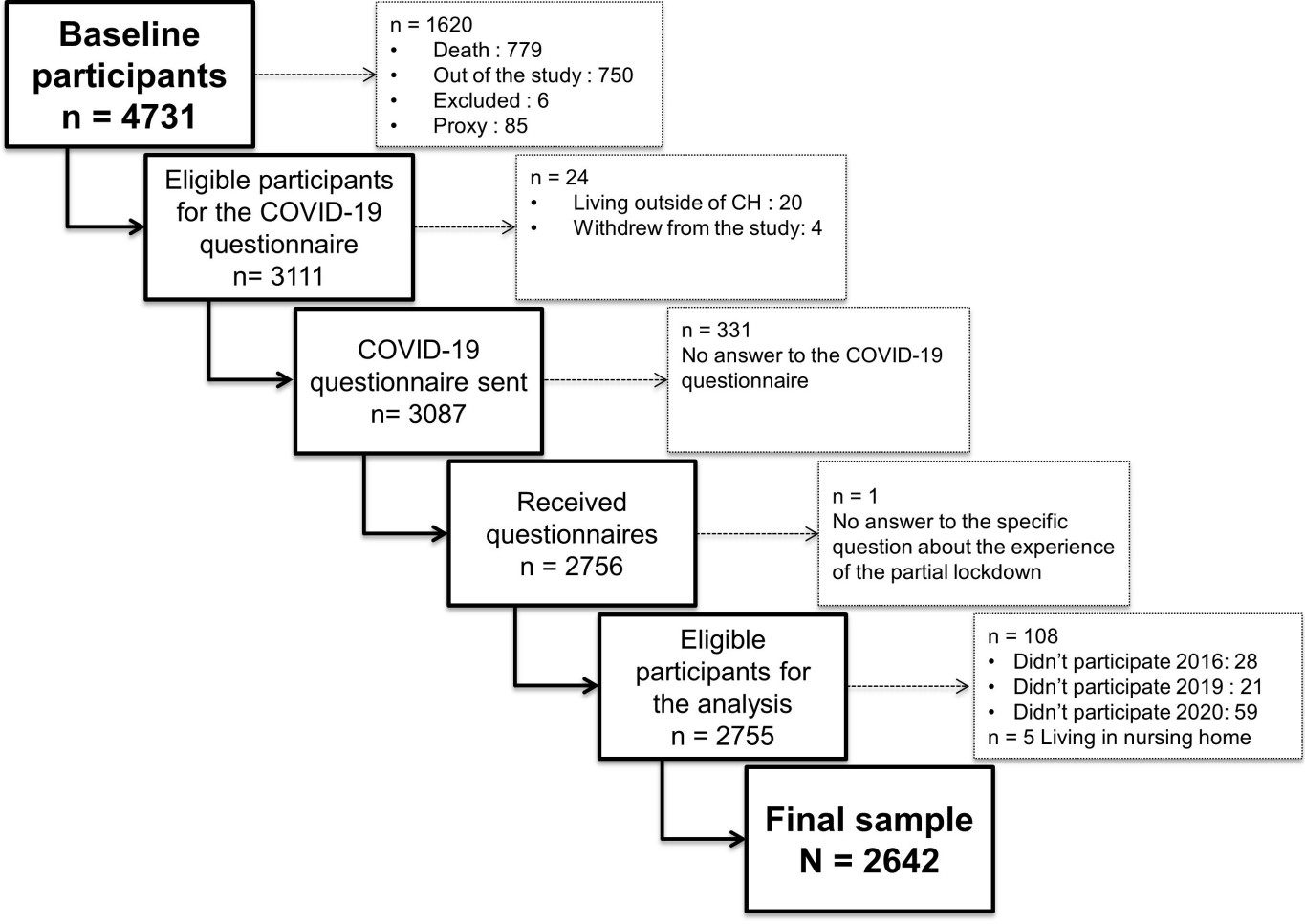

**Figure 1** Flow chart describing the selection of the study population.

school; medium: apprenticeship; high: college, university degree or equivalent) and financial difficulties (entitlement to supplementary benefits to old-age insurance).

### Living environment
Living environment variables encompassed living alone versus with other persons, home type (apartment vs house), home characteristics (having a balcony, a terrace/garden or none of the above) and home location (countryside/suburbs vs urban area).

### Health status
Health status variables included self-rated health, the presence or absence of comorbidities, impairment in basic and instrumental activities of daily living (BADL[18] and IADL[19]), mobility indicators (walking difficulties and fear of falling[20]), reporting memory difficulties or depressive/anxiety symptoms[21 22] and homebound status.

### Social status
Social status variables included usual social support,[23] emotional support,[24] feeling of loneliness, the recent death of a partner, being familiar with communication technology, participating in group activities and owning a dog and/or a cat.

Most information was retrieved from the 2019 questionnaire. If not available that year, data from the latest follow-up were used instead (for more details on timing of data collection for each variable, see online supplemental table 1 and figure 1).

### Statistical analysis
Proportions of participants reporting a positive, neutral and negative self-reported experience of the lockdown, respectively, were first compared in bivariable analyses within each characteristic of the study population, using parametric and non-parametric tests (ie, $\chi^2$ and Fisher exact tests) according to the variables' distribution. Multinomial logistic regression analyses were then performed to investigate the bivariable and multivariable associations between participants' characteristics and a positive or negative experience of the lockdown, using participants with neutral experience as the reference group. Variables that proved statistically significant ($p<0.05$) in bivariable analysis were included in the multivariable model.

Interactions were tested between gender and several specific characteristics (fear of falling, living alone, depressive and anxiety symptoms), as well as between living alone and usual social support, IADL, self-rated

health, emotional support or unfamiliarity with communication technologies.

To assess the effect of missing data, a sensitivity analysis was performed in a multivariate imputation approach. Assuming that covariables were missing at random, missing values were imputed iteratively by means of chained equations that generated 50 complete data sets. A multivariable analysis was run on each multiply imputed data set, which produced 50 sets of coefficients and SEs. The estimates were then pooled according to Rubin's combination rules.[25]

### Patient and public involvement

Patients and/or the public were not involved in the design, conduct, reporting, nor in the dissemination plans of this research.

### RESULTS

Overall, 2756 of the 3087 (response rate 89.3%) contacted participants returned their completed COVID-19 questionnaires and 2755 replied to the question about their experience of the lockdown. Figure 1 details the process of the population selection. After the exclusion of institutionalised participants (n=5), and of those who did not complete the previous questionnaires (n=108), analyses were conducted on 2642 participants (85.6% of sent questionnaires).

Compared with included participants (online supplemental table 2), those excluded (n=113) were older (p=0.002), more often women (p=0.033), had a lower education level (p<0.001), more frequently reported depressive symptoms (p<0.001), poor self-rated health (p=0.001) and a 'very difficult' or 'extremely difficult' experience of the lockdown (p=0.003).

Table 1 details participants' characteristics and provides the results of the comparisons across study groups according to their experience of the lockdown. Overall, 1791 (67.8%) participants reported a 'somewhat difficult' (neutral, reference) experience of the lockdown, 569 (21.5%) participants reported this experience as 'not at all difficult' (positive) and 282 (10.7%) as 'very or extremely difficult' (negative).

Missing values (n): education level (3), financial difficulties (1), living alone (8), home type (6), home location (22), home characteristics (5), comorbidities (13), BADL impairment (6), IADL impairment (36), walking difficulty (18), fear of falling (4), memory difficulties (12), depressive symptoms (3), anxiety symptoms (11), feelings of loneliness (4), usual social support (235), emotional support (15), previous participation in group activities (35), having a dog (4), having a cat (4), unfamiliarity with communication technology (30) and death of a partner in the past year (23).

Participants' experience of the lockdown did not differ according to their age (p=0.434) and education level (p=0.245). However, experience of the lockdown differed according to several other characteristics, some of which showed monotonically increasing or decreasing relationships, whereas others showed U-shaped or inverted U-shaped associations.

Among *sociodemographic* characteristics, women were less likely than men to report a positive experience of the lockdown (18.8% vs 25.7%), and more likely to report a negative one (11.9% vs 8.8%, p<0.001). By contrast, participants experiencing financial difficulties reported a negative experience more often than their peers did (13.2% vs 9.8%, p=0.035).

Most characteristics reflecting a comfortable *living environment* (eg, living in a house, having a terrace and/or garden) were also significantly associated with both higher rates of a positive experience and lower rates of a negative one (p<0.001). By contrast, living alone was associated with higher proportions of both a negative and a positive experience than living with others (p=0.001).

Similarly, most *health status* characteristics showed very important and significant intergroups differences. Participants with poor self-rated health (p<0.001), comorbidities (p=0.001), BADL impairment (p=0.013), IADL impairment (p<0.001), fear of falling (p<0.001), memory difficulties (p=0.002), depressive symptoms (p<0.001) and anxiety symptoms (p<0.001) reported more often a negative experience of the lockdown and less often a positive one than their counterparts.

Finally, *social status* characteristics also showed monotonic trends with a lower proportion of a positive experience and a higher proportion of a negative experience among participants feeling lonely (p<0.001), without emotional support (p=0.001), not having a dog (p=0.007) and unfamiliar with communication technology (p<0.001). The lack of social support (p=0.001) and previous participation in group activities (p=0.004) were associated with higher proportions of both negative and positive experiences and a lower proportion of a neutral one.

Table 2 shows the results of the *bivariate multinomial logistic regression*, comparing the groups reporting a positive experience and a negative experience to the neutral group (reference group).

The relative risk of reporting a *positive experience of the lockdown*—rather than a neutral one—was 34% *higher* among participants living alone (relative risk ratio, RRR 1.34, 95% CI 1.11 to 1.62), but 31% *lower* among women (RRR 0.69, 95% CI 0.57 to 0.84). It was also lower in participants with average, poor or very poor self-rated health, comorbidities, impairment in BADL and IADL, fear of falling, depressive symptoms, anxiety symptoms, a feeling of loneliness, previous participation in group activities or who were unfamiliar with communication technology (p<0.05 for all associations).

The relative risk of reporting a *negative experience of the lockdown*—rather than a neutral one—*increased* among participants who reported financial difficulties (RRR 1.43, 95% CI 1.09 to 1.88), but also in participants living

**Table 1** Characteristics of the study population and distributions by positive, neutral or negative experience of the lockdown

| | Experience of the lockdown* | | | | |
| --- | --- | --- | --- | --- | --- |
| | Total population n=2642 | Positive n=569 (21.5%) | Neutral (reference) n=1791 (67.8%) | Negative n=282 (10.7%) | P value† |
| Sociodemographic | | | | | |
| Age, mean (SD) | 78.0 (4.2) | 77.8 (4.2) | 78.1 (4.1) | 78.1 (4.4) | 0.434 |
| Gender, n (%) | | | | | |
| Men | 1059 (100.0) | 272 (25.7) | 694 (65.5) | 93 (8.8) | <0.001 |
| Women | 1583 (100.0) | 297 (18.8) | 1097 (69.3) | 189 (11.9) | |
| Education level, n (%) | | | | | |
| Basic compulsory | 404 (100.0) | 89 (22.0) | 261 (64.6) | 54 (13.4) | 0.245 |
| Apprenticeship | 1032 (100.0) | 210 (20.4) | 712 (69.0) | 110 (10.7) | |
| Postcompulsory schooling | 1203 (100.0) | 268 (22.3) | 817 (67.9) | 118 (9.8) | |
| Financial difficulties, n (%) | | | | | |
| No | 1960 (100.0) | 420 (21.4) | 1348 (68.8) | 192 (9.8) | 0.035 |
| Yes | 681 (100.0) | 149 (21.9) | 442 (64.9) | 90 (13.2) | |
| Living environment | | | | | |
| Living alone, n (%) | | | | | |
| No | 1465 (100.0) | 288 (19.7) | 1037 (70.8) | 140 (9.6) | 0.001 |
| Yes | 1169 (100.0) | 279 (23.9) | 748 (64.0) | 142 (12.1) | |
| Home type, n (%) | | | | | |
| Apartment | 2298 (100.0) | 475 (20.7) | 1556 (67.7) | 267 (11.6) | <0.001 |
| House | 338 (100.0) | 91 (26.9) | 232 (68.6) | 15 (4.4) | |
| Home location, n (%) | | | | | |
| City | 2061 (100.0) | 428 (20.8) | 1401 (68.0) | 232 (11.3) | 0.052 |
| Suburbs/rural areas | 559 (100.0) | 134 (24.0) | 379 (67.8) | 46 (8.2) | |
| Home characteristics, n (%) | | | | | |
| No balcony/terrace/garden | 140 (100.0) | 29 (20.7) | 98 (70.0) | 13 (9.3) | <0.001 |
| A balcony | 1684 (100.0) | 339 (20.1) | 1128 (67.0) | 217 (12.9) | |
| A terrace and/or garden | 813 (100.0) | 198 (24.4) | 563 (69.3) | 52 (6.4) | |
| Health status | | | | | |
| Self-rated health, n (%) | | | | | |
| Very good/good | 1734 (100.0) | 419 (24.2) | 1173 (67.6) | 142 (8.2) | <0.001 |
| Average/poor/very poor | 908 (100.0) | 150 (16.5) | 618 (68.1) | 140 (15.4) | |
| Comorbidities, n (%) | | | | | |
| None | 1150 (100.0) | 288 (25.0) | 760 (66.1) | 102 (8.9) | 0.001 |
| 1 | 981 (100.0) | 189 (19.3) | 675 (68.8) | 117 (11.9) | |
| 2+ | 498 (100.0) | 87 (17.5) | 350 (70.3) | 61 (12.3) | |
| BADL impairment,‡ n (%) | | | | | |
| Independent | 2172 (100.0) | 486 (22.4) | 1467 (67.5) | 219 (10.1) | 0.013 |
| Difficulty or help for at least 1 | 464 (100.0) | 80 (17.2) | 322 (69.4) | 62 (13.4) | |
| IADL impairment,§ n (%) | | | | | |
| Independent | 1239 (100.0) | 329 (26.6) | 819 (66.1) | 91 (7.3) | <0.001 |
| Difficulty or help for at least 1 | 1367 (100.0) | 225 (16.5) | 954 (69.8) | 188 (13.8) | |
| Walking difficulty, n (%) | | | | | |
| No | 2285 (100.0) | 493 (21.6) | 1554 (68.0) | 238 (10.4) | 0.546 |
| Yes | 339 (100.0) | 71 (20.9) | 226 (66.7) | 42 (12.4) | |
| Fear of falling, n (%) | | | | | |
| No | 1136 (100.0) | 316 (27.8) | 744 (65.5) | 76 (6.7) | <0.001 |

Continued

**Table 1** Continued

| | Experience of the lockdown* | | | | |
| --- | --- | --- | --- | --- | --- |
| | Total population n=2642 | Positive n=569 (21.5%) | Neutral (reference) n=1791 (67.8%) | Negative n=282 (10.7%) | P value† |
| Yes | 1502 (100.0) | 253 (16.8) | 1044 (69.5) | 205 (13.7) | |
| Homebound status, n (%) | | | | | |
| Non-homebound | 2046 (100.0) | 437 (21.4) | 1387 (67.8) | 222 (10.9) | 0.552 |
| Semi-homebound | 577 (100.0) | 125 (21.7) | 394 (68.3) | 58 (10.1) | |
| Homebound | 19 (100.0) | 7 (36.8) | 10 (52.6) | 2 (10.5) | |
| Memory difficulties, n (%) | | | | | |
| No | 2218 (100.0) | 492 (22.2) | 1507 (67.9) | 229 (9.9) | 0.002 |
| Yes | 412 (100.0) | 74 (18.0) | 275 (66.7) | 63 (15.3) | |
| Depressive symptoms, n (%) | | | | | |
| No | 1916 (100.0) | 460 (24.0) | 1302 (68.0) | 154 (8.0) | <0.001 |
| Yes | 723 (100.0) | 108 (14.9) | 487 (67.4) | 128 (17.7) | |
| Anxiety symptoms, n (%) | | | | | |
| No | 1693 (100.0) | 423 (25.0) | 1130 (66.8) | 140 (8.3) | <0.001 |
| Yes | 938 (100.0) | 143 (15.3) | 653 (69.6) | 142 (15.1) | |
| Social status | | | | | |
| Feelings of loneliness, n (%) | | | | | |
| No | 1954 (100.0) | 472 (24.2) | 1311 (67.1) | 171 (8.8) | <0.001 |
| Yes | 684 (100.0) | 95 (13.9) | 479 (70.0) | 110 (16.1) | |
| Usual social support, n (%) | | | | | |
| No | 596 (100.0) | 138 (23.2) | 374 (62.8) | 84 (14.1) | 0.001 |
| Yes | 1811 (100.0) | 378 (20.9) | 1266 (69.9) | 167 (9.2) | |
| Emotional support, n (%) | | | | | |
| 0 | 255 (100.0) | 51 (20.0) | 162 (63.5) | 42 (16.5) | 0.001 |
| 1–2 | 398 (100.0) | 74 (18.6) | 269 (67.6) | 55 (13.8) | |
| 3 | 1974 (100.0) | 440 (22.3) | 1351 (68.4) | 183 (9.3) | |
| Previous participation in group activities, n (%) | | | | | |
| No | 1112 (100.0) | 263 (23.7) | 716 (64.4) | 133 (12.0) | 0.004 |
| Yes | 1495 (100.0) | 297 (19.9) | 1054 (70.5) | 144 (9.6) | |
| Having a dog, n (%) | | | | | |
| No | 2508 (100.0) | 533 (21.3) | 1697 (67.7) | 278 (11.1) | 0.007 |
| Yes | 130 (100.0) | 36 (27.7) | 90 (69.2) | 4 (3.1) | |
| Having a cat, n (%) | | | | | |
| No | 2301 (100.0) | 488 (21.2) | 1577 (68.5) | 236 (10.3) | 0.051 |
| Yes | 337 (100.0) | 81 (24.0) | 210 (62.3) | 46 (13.6) | |
| Unfamiliarity with communication technology, n (%) | | | | | |
| No | 1883 (100.0) | 458 (24.3) | 1252 (66.5) | 173 (9.2) | <0.001 |
| Yes | 729 (100.0) | 105 (14.4) | 517 (70.9) | 107 (14.7) | |
| Death of a partner in the past year, n (%) | | | | | |
| No | 2561 (100.0) | 550 (21.5) | 1735 (67.7) | 276 (10.8) | 0.607 |
| Yes | 58 (100.0) | 12 (20.7) | 42 (72.4) | 4 (6.9) | |

*Self-reported experience defined as 'positive', 'neutral' and 'negative' in participants reporting having experienced the lockdown as 'not at all difficult', 'somewhat difficult' and 'very or extremely difficult', respectively.
†P values from ANOVA test for age, and from Pearson's $\chi^2$-test or Fischer exact test when expected frequencies <5, for categorical variables.
‡BADL, basic activities of daily living (bathing, dressing, toileting, transferring, feeding and continence).
§IADL, instrumental activities of daily living (ability to use the telephone, to use transportations, prepare food, groceries, do the housekeeping, handle medications and manage finances).
ANOVA, analysis of variance.

**Table 2** Results from bivariable multinomial logistic regression investigating participants' characteristics associated with a positive ('not at all difficult') and a negative ('very or extremely difficult') experience of the lockdown, using participants with a neutral ('slightly difficult') experience as reference group

| | Experience of the lockdown | | | | | |
| --- | --- | --- | --- | --- | --- | --- |
| | Positive (ref. neutral) | | | Negative (ref. neutral) | | |
| | RRR | 95% CI | P value* | RRR | 95% CI | P value* |
| Sociodemographic | | | | | | |
| Age | 0.99 | 0.96 to 1.01 | 0.223 | 1.00 | 0.97 to 1.03 | 0.840 |
| Gender | | | | | | |
| Men | Ref. | | | Ref. | | |
| Women | 0.69 | 0.57 to 0.84 | <0.001 | 1.29 | 0.99 to 1.68 | 0.064 |
| Education level | | | | | | |
| Basic compulsory | Ref. | | | Ref. | | |
| Apprenticeship | 0.86 | 0.65 to 1.15 | 0.319 | 0.75 | 0.52 to 1.07 | 0.107 |
| Postcompulsory schooling | 0.96 | 0.73 to 1.27 | 0.784 | 0.70 | 0.49 to 0.99 | 0.045 |
| Financial difficulties | | | | | | |
| No | Ref. | | | Ref. | | |
| Yes | 1.08 | 0.87 to 1.34 | 0.474 | 1.43 | 1.09 to 1.88 | 0.010 |
| Living environment | | | | | | |
| Living alone | | | | | | |
| No | Ref. | | | Ref. | | |
| Yes | 1.34 | 1.11 to 1.62 | 0.002 | 1.41 | 1.09 to 1.81 | 0.008 |
| Home type | | | | | | |
| Apartment | Ref. | | | Ref. | | |
| House | 1.28 | 0.99 to 1.67 | 0.062 | 0.38 | 0.22 to 0.65 | <0.001 |
| Home location | | | | | | |
| City | Ref. | | | Ref. | | |
| Suburbs/rural areas | 1.16 | 0.92 to 1.45 | 0.203 | 0.73 | 0.52 to 1.03 | 0.070 |
| Home characteristics | | | | | | |
| No balcony, terrace or garden | 0.98 | 0.64 to 1.52 | 0.944 | 0.69 | 0.38 to 1.25 | 0.222 |
| A balcony | Ref. | | | Ref. | | |
| A terrace and/or garden | 1.17 | 0.96 to 1.43 | 0.128 | 0.48 | 0.35 to 0.66 | <0.001 |
| Health status | | | | | | |
| Self-rated health | | | | | | |
| Very good/good | Ref. | | | Ref. | | |
| Average/poor/very poor | 0.68 | 0.55 to 0.84 | <0.001 | 1.87 | 1.45 to 2.41 | <0.001 |
| Comorbidities | | | | | | |
| None | Ref. | | | Ref. | | |
| 1 | 0.74 | 0.60 to 0.91 | 0.005 | 1.29 | 0.97 to 1.72 | 0.079 |
| 2+ | 0.66 | 0.50–0.86 | 0.002 | 1.30 | 0.92–1.83 | 0.134 |
| BADL impairment | | | | | | |
| Independent | Ref. | | | Ref. | | |
| Diff./help for at least 1 | 0.75 | 0.58–0.98 | 0.034 | 1.29 | 0.95–1.75 | 0.104 |
| IADL impairment | | | | | | |
| Independent | Ref. | | | Ref. | | |
| Diff./help for at least 1 | 0.59 | 0.48–0.71 | <0.001 | 1.77 | 1.36–2.32 | <0.001 |
| Walking difficulty | | | | | | |
| No | Ref. | | | Ref. | | |
| Yes | 0.99 | 0.74–1.32 | 0.946 | 1.21 | 0.85–1.73 | 0.287 |
| Fear of falling | | | | | | |

Continued

**Table 2** Continued

| | Experience of the lockdown | | | | | |
| | Positive (ref. neutral) | | | Negative (ref. neutral) | | |
| | RRR | 95% CI | P value* | RRR | 95% CI | P value* |
|---|---|---|---|---|---|---|
| No | Ref. | | | Ref. | | |
| Yes | 0.57 | 0.47–0.69 | <0.001 | 1.92 | 1.45–2.54 | <0.001 |
| Homebound status | | | | | | |
| Non-homebound | Ref. | | | Ref. | | |
| Semi-homebound | 1.01 | 0.80–1.26 | 0.953 | 0.92 | 0.67–1.25 | 0.597 |
| Homebound | 2.22 | 0.84–5.87 | 0.107 | 1.25 | 0.27–5.74 | 0.775 |
| Memory difficulties | | | | | | |
| No | Ref. | | | Ref. | | |
| Yes | 0.82 | 0.63–1.09 | 0.170 | 1.58 | 1.16–2.15 | 0.004 |
| Depressive symptoms | | | | | | |
| No | Ref. | | | Ref. | | |
| Yes | 0.63 | 0.50–0.79 | <0.001 | 2.22 | 1.72–2.87 | <0.001 |
| Anxiety symptoms | | | | | | |
| No | Ref. | | | Ref. | | |
| Yes | 0.59 | 0.47–0.72 | <0.001 | 1.76 | 1.36–2.26 | <0.001 |
| Social status | | | | | | |
| Feelings of loneliness | | | | | | |
| No | Ref. | | | Ref. | | |
| Yes | 0.55 | 0.43–0.70 | <0.001 | 1.76 | 1.36–2.29 | <0.001 |
| Usual social support | | | | | | |
| No | Ref. | | | Ref. | | |
| Yes | 0.81 | 0.65–1.01 | 0.067 | 0.59 | 0.44–0.78 | <0.001 |
| Emotional support | | | | | | |
| 0 | Ref. | | | Ref. | | |
| 1–2 | 0.87 | 0.58–1.31 | .515 | 0.79 | 0.50–1.23 | 0.297 |
| 3 | 1.03 | 0.74–1.44 | 0.841 | 0.52 | 0.36–0.76 | <0.001 |
| Previous participation in group activities | | | | | | |
| No | Ref. | | | Ref. | | |
| Yes | 0.77 | 0.63–0.93 | 0.007 | 0.74 | 0.57–0.95 | 0.018 |
| Having a dog | | | | | | |
| No | Ref. | | | Ref. | | |
| Yes | 1.27 | 0.85–1.90 | 0.234 | 0.27 | 0.10–0.74 | 0.011 |
| Having a cat | | | | | | |
| No | Ref. | | | Ref. | | |
| Yes | 1.25 | 0.95–1.64 | 0.117 | 1.46 | 1.03–2.07 | 0.031 |
| Unfamiliarity with communication technology | | | | | | |
| No | Ref. | | | Ref. | | |
| Yes | 0.56 | 0.44–0.70 | <0.001 | 1.50 | 1.15–1.95 | 0.003 |
| Death of a partner in the past year | | | | | | |
| No | Ref. | | | Ref. | | |
| Yes | 0.90 | 0.47–1.72 | 0.754 | 0.60 | 0.21–1.68 | 0.331 |

*P-values from multinomial logistic regression.
RRR, relative risk ratio.

alone, who rated their health as average, poor or very poor, reported IADL impairment, fear of falling, memory difficulties, depressive symptoms, anxiety symptoms, feelings of loneliness, having a cat and being unfamiliar with communication technology (p<0.05 for all associations). On the contrary, the relative risk *decreased* among participants with a postcompulsory education level (RRR 0.70, 95% CI 0.49 to 0.99), but also in participants who lived in a house, had access to a terrace and/or garden, benefited from usual social support, reported at least three sources of emotional support, had previously participated in group activities and owned a dog (p<0.05 for all associations).

Interestingly, some characteristics showed significant associations in a consistent direction, with increased relative risks of a negative experience, on the one hand, and decreased relative risks of a positive experience on the other hand. For instance, participants who reported fear of falling had a 43% lower relative risk of reporting a positive experience (RRR 0.57, 95% CI 0.47 to 0.69) and were almost twice more likely to report a negative experience (RRR 1.92, 95% CI 1.45 to 2.54) of the lockdown as those without any fear of falling. Similar associations with both significant positive and negative experience were also observed for self-rated health, IADL impairment, depressive symptoms, anxiety symptoms, feelings of loneliness and being unfamiliar with communication technology.

Two characteristics showed a U-shaped association with the experience of the lockdown. Participants living alone had a higher relative risk to report both a positive (RRR 1.34, 95% CI 1.11 to 1.62) and a negative (RRR 1.41, 95% CI 1.09 to 1.81) experience rather than a neutral one. Inversely, previous participation in group activities (reported by 59.5% of the total population) was associated with decreased relative risks of reporting a positive (RRR 0.77, 95% CI 0.63 to 0.93) or a negative (RRR 0.74, 95% CI 0.57 to 0.95) experience of the lockdown.

Figure 2 shows the results of the *multivariable multinomial logistic regression* (for more details, see online supplemental table 3). The relative risk of reporting a *positive experience*—rather than a neutral one—remained *higher* for participants who lived alone (RRR 1.93, 95% CI 1.52 to 2.46) or in a house (RRR 1.49, 95% CI 1.03 to 2.16), and *lower* for women (RRR 0.75, 95% CI 0.59 to 0.95), for participants who reported IADL impairment (RRR 0.78, 95% CI 0.61 to 0.99), fear of falling (RRR 0.68, 95% CI 0.54 to 0.86), feelings of loneliness (RRR 0.67, 95% CI 0.49 to 0.91), for those who received usual social support (RRR 0.71, 95% CI 0.54 to 0.93) and for those who had previously participated in group activities (RRR 0.74, 95% CI 0.59 to 0.92) or were unfamiliar with communication technology (RRR 0.69, 95% CI 0.52 to 0.91).

Few characteristics remained independently associated with a *negative experience* of the lockdown. The relative risk of the latter—compared with a neutral one—*increased* among participants with fear of falling (RRR 1.52, 95% CI 1.07 to 2.15), and *decreased* among those who had access to a terrace and/or garden (RRR 0.66, 95% CI 0.44 to 0.99) or owned a dog (RRR 0.32, 95% CI 0.11 to 0.9).

In the multivariable model, fear of falling was the only factor that remained significantly associated in *a consistent direction*, with an increased relative risk of a negative experience (RRR 1.52, 95% CI 1.07 to 2.15) and a decreased relative risk of a positive experience (RRR 0.68, 95% CI 0.54 to 0.86) compared with the neutral group.

None of the tested interactions were significant. The sensitivity analysis conducted using the multivariate imputation approach produced very similar results (see online supplemental table 3).

## DISCUSSION

This study examined the experience of the COVID-19 partial lockdown in a community-dwelling Swiss older population. Its unique contribution was to highlight which participants' characteristics were independently associated with a positive and a negative experience, respectively. Results confirmed our initial hypothesis that most participants (ie, about two-thirds) would report a neutral experience of the lockdown and describe this period as 'somewhat difficult'. By contrast, only about one in 10 older people reported a negative experience of the lockdown, describing it as 'very' or 'extremely' difficult. At the other end of the spectrum, more than a fifth of the participants reported a positive experience and described the lockdown as 'not difficult at all'. Most of the observed associations were in line with our hypotheses: participants who lived in a more spacious environment (living in a house) and/or with outdoor access (having a terrace and/or garden) were significantly less likely than their counterparts to report a negative rather than a neutral experience. Reversely, participants with health problems (fear of falling and functional difficulties) as well as those who were socially vulnerable (who reported feelings of loneliness, receiving usual social support and being unfamiliar with communication technology) were significantly less likely to report a positive experience.

Several notable associations were rather unexpected. For instance, contrary to our initial hypothesis, living alone turned out to be associated with a higher relative risk of a positive experience. This result contrasts with another study that identified living alone as a vulnerability factor during a lockdown.[26] The possibility to adjust for the feeling of loneliness might explain the current study's finding in that regard. Indeed, participants who expressed feelings of loneliness were significantly less likely than their counterparts to report a positive rather than a neutral experience—a finding that is consistent with the results of another Swiss study.[14] This hypothesis is further supported by the observation that participants who had usual social support and had previously participated in group activities were more likely to report a negative experience of the lockdown. Overall, these results suggest that older participants who were socially more active and involved, and those who already felt lonely

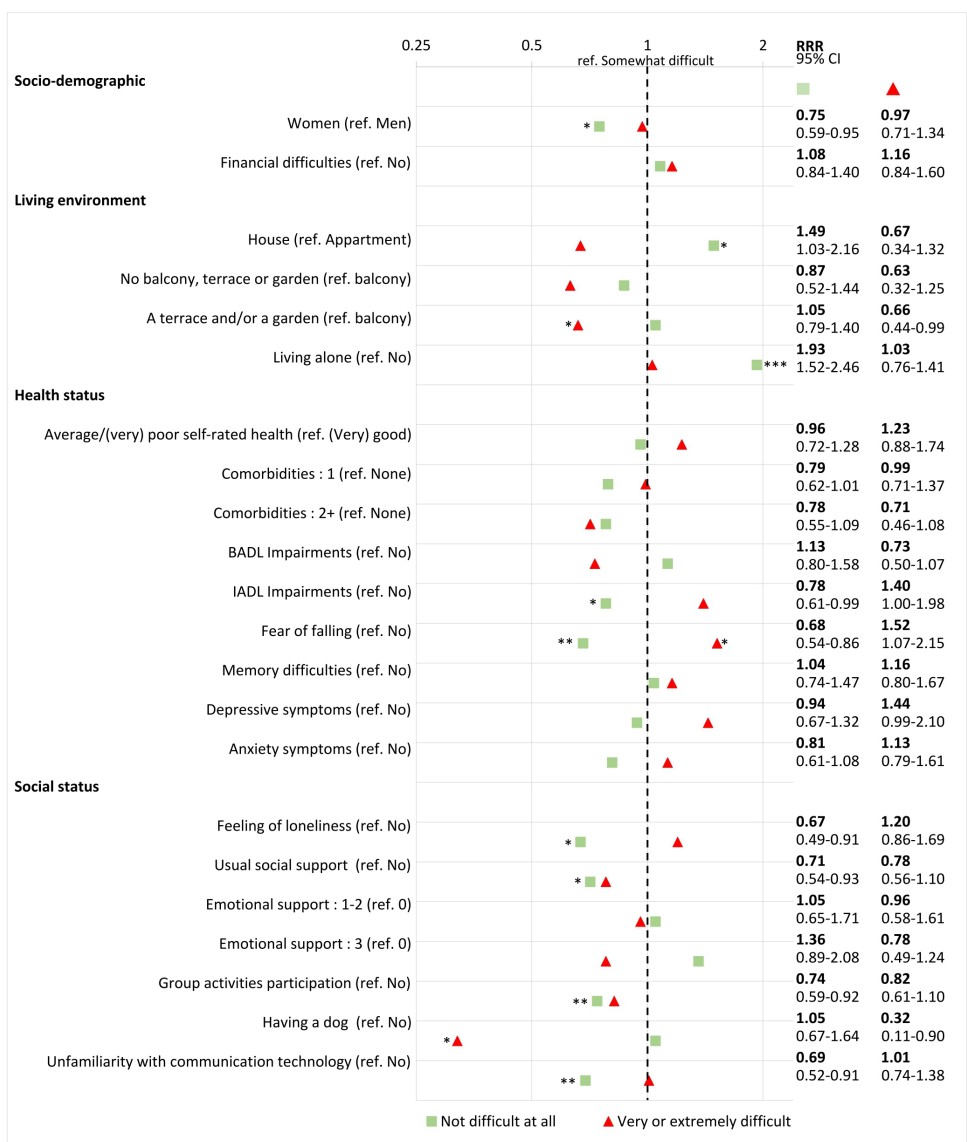

**Figure 2** Results from the multivariable multinomial logistic regression analysis investigating participants' characteristics associated with a self-reported positive (green boxes, 'not difficult at all') and negative (red triangles, 'very or extremely difficult') experience of the lockdown, using participants with neutral ('slightly difficult') experience as a reference group. Relative risk ratio (RRR) from the multivariable multinomial logistic regression. ***P<0.001, **p<0.01, *p<0.05. BADL, basic activities of daily living; IADL, instrumental activities of daily living.

were less able to cope with the lockdown than those who were already living alone before the pandemic. Indeed, older people who still live alone in their home tend to be more mobile, more independent and autonomous and may not be particularly prone to social isolation.[27] However, this hypothesis is not supported in this study. Indeed, 36.3% of participants who lived alone reported feelings of loneliness, compared with 17.6% of those who lived together with other people. In this regard, the observed relationship between participants' unfamiliarity with communication technology and a reduced chance of reporting a positive experience of the lockdown is particularly interesting. Keeping in touch with their social network during the lockdown was crucial for older people, as some reported feeling cut-off from society.[28] Communication technology was identified as a coping

strategy in hospitals and nursing homes. Health teams often used the technology of tablets to help older people keep in touch with their families in times when visits were forbidden. Results from the current study extend current knowledge as findings about the role of information technology and Internet access in preventing older adults from feeling lonely and socially disconnected are discordant.[29–31] Our observation provides further evidence of the need to foster the training and supervision of older people in using communication technology to reduce their loneliness and social isolation.

The lack of association between participants' age and their experience of the lockdown was also contrary to our hypothesis. As some studies suggest, the relationship between age and coping mechanisms might be more complex, as older people tend to show more resilience

than younger ones during stressful times.[12] Similarly, the lack of association between a partner's death in the past year and the experience of the lockdown might be surprising. However, such a lack likely resulted from the low (2.2%) incidence of widowhood in the study population. Unfortunately, the lack of data on the specific timing and cause of death made it impossible to investigate the likelihood of an association in a subgroup whose spouse died recently, and/or from a SARS-CoV-2 infection.

Interestingly, this study further contributed to highlighting the independent association between the experience of the lockdown and several characteristics of the participants' living environment. Although none of these characteristics showed a significant independent association in a consistent direction (ie, a decreased risk of a negative experience and an increased chance of a positive experience or reversely), living in a house and having access to a terrace and/or garden appeared to influence the experience of the lockdown in one of the hypothesised directions. Fortunately, few participants (5.3%) had no access to a balcony, a terrace and/or garden—a factor worth examining in case of another lockdown in the future.

The robust observation of the association between the participants' fear of falling and their experience of the lockdown is meaningful, too. Indeed, fear of falling was the only variable that remained independently associated in a consistent direction: fearful participants were both at an increased relative risk of reporting a negative experience and at a lower relative risk of reporting a positive experience. Notably, this association remained, even after extensive adjustment for other measurements of the health status (including anxiety), as well as for sociodemographic, living environment and social status characteristics. The reason for this may be that the fear of falling reflects a state of increased combined mental and physical vulnerability, as also suggested by other studies that found a robust association with outdoor mobility and quality of life in community-dwelling older people.[32–34] Further research should investigate whether the fear of falling should be addressed through interventions, such as exercise, primary fall prevention, as well as behavioural and cognitive therapies.[35]

Finally, the observation that dog owners were particularly unlikely to report a negative experience, whereas cat owners were likely to do so, deserves a comment. This suggests that the relationship between dog ownership and the experience of the lockdown occurred through enhanced mobility rather than merely through possessing a pet animal. This goes in line with a recent Japanese study,[36] which showed an association between dog ownership and a reduced risk of disability, possibly because of the resulting moderate physical activity maintained. Alternative explanations could also be provided by previously reported differences between cat and dog owners in feelings of loneliness and social isolation, as well as by differences in personality patterns.[37 38] To encourage older people to practise dog walking could be an interesting option in times of a lockdown—and probably also in general—as it might prevent them from developing disability and frailty, but also help them enjoy more social contact and reduce feelings of loneliness.

## Strengths and limitations

A major strength of this population-based study is the use of postal—rather than online—questionnaires, for it allowed to include the subgroup of individuals who are uncomfortable with the use of communication technology. The fact that participants completed the questionnaires during the lockdown excludes any memory bias. Furthermore, the design of the study allowed to link the data collected during the lockdown with key characteristics routinely assessed in the Lc65+ study.

This study has certain limitations. First, because the results were obtained from community-dwelling adults aged 71–86 years, they cannot be generalised to all older individuals. Second, the most vulnerable participants had to be excluded from the study due to missing data or institutionalisation. Fortunately, excluded individuals represented only a very low percentage of the total sample (7.5%). In terms of external validity, this study focused on a partial lockdown. During the same period, many other countries implemented a stricter or even a total lockdown. Furthermore, this study focused on the specific part of the older population that lives in the community, is in relatively good health and is relatively independent. Accordingly, it might be that the partial lockdown was less restrictive for this category of older people regarding social interactions than for older people who live in a nursing home or who visit day-care facilities. Finally, data on sociodemographic, living environment, health and social characteristics stem mainly from the 2019 questionnaire. Although living environment and sociodemographic characteristics do not appear particularly prone to substantial variations from 1 year to another, we cannot formally exclude minor changes in health and social characteristics from 2019 to 2020. Nevertheless, if preventive interventions are to be implemented in the future to limit the impact of stringent distancing measures, data available to target those interventions will have been collected before the introduction of the measures for a timely and efficient approach.

## Future implications

This study identified that many participants' characteristics regarding sociodemography, living environment, health and social status were related with their experience of the lockdown. From a public health perspective, this study further highlights the importance of targeted interventions in reducing the impact of another potential lockdown in the future. For example, maintaining a higher level of mobility could be achieved by actions such as accompanying older people for outside walks or more specifically by means of actions that alleviate the fear of falling (eg, exercise, behavioural and cognitive therapy) even if such interventions must stay in line with sanitary

measures and individuals' security. Modifying the physical environment and promoting activities such as walking a dog could also be recommended to address social isolation. Finally, this study's findings add to previous evidence of the necessary empowerment of older people, especially of those who express feelings of loneliness; by learning how to master and use communication technology, benefit from more social and emotional support and keep in touch with their relatives.

**Acknowledgements** We sincerely thank all participants of the Lc65+ study. We are also grateful to the Lc65+ collaborators for their involvement in the data collection, as well as to Professor Brigitte Santos-Eggimann for her contribution to the conception of the COVID-19 specific questionnaire.

**Contributors** DM-G drafted the manuscript, SF conducted the analyses, YH supervised the work, designed and conducted the data collection as the principal investigator of the Lc65+ study, accepts full responsibility of the work and/or the conduct of the study, had access to the data and controlled the decision to publish. CB and LS-B designed and supervised the work. All authors (DM-G, SF, YH, CB, LS-B and JS) read, revised and approved the final manuscript.

**Funding** From the beginning, the Lc65+ study has been financed exclusively by public funds or not-for-profit organisations. It is currently funded by the Public Health Department of canton Vaud, the Centre for Primary Care and Public Health (Unisanté) and by research grants from the Loterie Romande (not-for-profit organisation supporting research and social projects) and the Esther Locher-Gurtner Foundation.

**Competing interests** None declared.

**Patient and public involvement** Patients and/or the public were not involved in the design, or conduct, or reporting or dissemination plans of this research.

**Patient consent for publication** Not required.

**Ethics approval** The Cantonal Human Research Ethical Committee approved the Lausanne Cohort Lc65+ initial study protocol (Protocol (19/04), decision letter: 23 February 2004), the successive amendments and the COVID-19 questionnaire for follow-up. All participants gave their written informed consent.

**Provenance and peer review** Not commissioned; externally peer reviewed.

**Data availability statement** All data relevant to the study are included in the article or uploaded as supplementary information.

**ORCID iD**
Daphné Märki-Germann http://orcid.org/0000-0003-2159-367X

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
