## [Reviewer comments · BMJ Open]

ARTICLE DETAILS

TITLE (PROVISIONAL)	« Older people's experience of the partial lockdown during the COVID-19 pandemic in Switzerland: a cross-sectional study »
AUTHORS	Märki-Germann, Daphné; Seematter-Bagnoud, Laurence; FUSTINONI, Sarah; SPALTENSTEIN, Julia; Bula, Christophe; HENCHOZ, Yves

VERSION 1 – REVIEW

REVIEWER	Cadmus, Eniola University of Ibadan, Community Medicine
REVIEW RETURNED	06-Sep-2022

GENERAL COMMENTS	Abstract conclusion. Authors should specify what specific intervention considering the study findings. Confusing statement in the results section of the abstract The relative risk of a positive experience was lower in those who reported their fear of falling (RRR=0.68, 0.54–0.86) It seems similar to the statement later on 'the relative risk of a negative experience was higher in participants with fear of falling (RRR=1.52, 1.07–2.15)'. Kindly clarify
---

REVIEWER	Fadda, Marta Università della Svizzera italiana, Institute of Communication and Health
REVIEW RETURNED	07-Oct-2022

GENERAL COMMENTS	Thank you for the opportunity to review this manuscript. I have some minor comments: - "Our main hypothesis was that most participants would report a neutral feeling, judging this period as somewhat difficult, whereas two smaller subgroups would report more extreme feelings and consider their experience as either negative (judging the period as very difficult) or positive (judging the period as not difficult at all)." How did you develop this hypothesis? An explanation is provided for your second hypothesis, but not for such distribution that you expected the data to reveal.- "Most information was retrieved from the 2019 questionnaire." How did you ensure that the situation was still similar in 2020, when you administered the lockdown-related questionnaire?- "The robust observation of the association between participants' fear of falling and their experience of the lockdown is meaningful, too." I think this is a very interesting result, that shows well how
---

	frailty is a very complex concept that may be conceived not only as an outcome but also as a driver (in this case, determining a certain lockdown experience). I would like the authors to unpack this finding more and provide more details on the practical implications that addressing frailty has for public health purposes.
REVIEWER	Kumar, Anuj Apeejay School of Management
REVIEW RETURNED	05-Jan-2023
GENERAL COMMENTS	Manuscript is acceptable

VERSION 1 – AUTHOR RESPONSE

Reviewer: 1
Dr. Eniola Cadmus, University of Ibadan

Comments to the Author:

-Abstract conclusion.
Authors should specify what specific intervention considering the study findings.

Response:
Thank you for the review. We added in the last sentence a statement referring to interventions addressing the most potent risk factor identified in the study, i.e., fear of falling.

-Confusing statement in the results section of the abstract
The relative risk of a positive experience was lower in those who reported their fear of falling (RRR=0.68, 0.54–0.86)
It seems similar to the statement later on 'the relative risk of a negative experience was higher in participants with fear of falling (RRR=1.52, 1.07–2.15)'. Kindly clarify

Response:
It is true that this result can seem confusing at first glance. However, as previously stated (please see our response to the editor's last comment), a variable significantly associated with a negative experience has not necessarily also an inverse significant association with a positive experience. Indeed, fear of falling is the only variable that was significantly associated with the lockdown experience in two congruent directions:
The relative risk of a positive experience (compared to a neutral one) was lower in participants who reported a fear of falling AND, at the same time, these participants also had a significantly higher relative risk of a negative experience (compared to a neutral one). These results are explained in Figure 2.

Reviewer: 2
Dr. Marta Fadda, Università della Svizzera italiana

Comments to the Author:
Thank you for the opportunity to review this manuscript. I have some minor comments:

- "Our main hypothesis was that most participants would report a neutral feeling, judging this period as somewhat difficult, whereas two smaller subgroups would report more extreme feelings and consider their experience as either negative (judging the period as very difficult) or positive (judging the period as not difficult at all)." How did you develop this hypothesis? An explanation is provided for your second hypothesis, but not for such distribution that you expected the data to reveal.

Response:

Thank you for your review. At the time of building this hypothesis, no scientific evidence was available to guide the research team's discussions, that were therefore mainly based on the Swiss context during this unprecedented situation. We added the following sentence to support our main hypothesis: "Since social distancing measures particularly targeted the older population, but the lockdown was less strict in Switzerland than in many other countries, our hypothesis was that most participants would report a neutral experience [...]"

- "Most information was retrieved from the 2019 questionnaire." How did you ensure that the situation was still similar in 2020, when you administered the lockdown-related questionnaire?

Response:

Thank you for this very relevant remark. Although most socio-demographic, living environment, health, and social factors considered in this study have been generally quite stable from one year to another in our cohort (e.g. education level, financial difficulties, home characteristics, comorbidities, social support, etc), we cannot formally exclude possible minor changes over time in some specific characteristic, which we could unfortunately not measure. The best we could do was to choose these variables as close as possible to March 2020. We then included this comment in our study limitations: "Finally, data on socio-demographic, living environment, health, and social characteristics stem mainly from the 2019 questionnaire. Although living environment and socio-demographic characteristics do not appear particularly prone to substantial variations from one year to another, we cannot formally exclude minor changes in health and social characteristics from 2019 to 2020. Nevertheless, if preventive interventions are to be implemented in the future to limit the impact of stringent distancing measures, data available to target those interventions will have been collected before the introduction of the measures for a timely and efficient approach."

- "The robust observation of the association between participants' fear of falling and their experience of the lockdown is meaningful, too." I think this is a very interesting result, that shows well how frailty is a very complex concept that may be conceived not only as an outcome but also as a driver (in this case, determining a certain lockdown experience). I would like the authors to unpack this finding more and provide more details on the practical implications that addressing frailty has for public health purposes.

Response:

Thank you for this comment. Accordingly, we completed a paragraph in the discussion with additional details:

"Further research should investigate whether fear of falling should be addressed through interventions, such as exercise, primary fall prevention, as well as behavioral and cognitive therapies." We also provided an additional reference to a Cochrane systematic review concerning fear of falling. MacKay S, Ebert P, Harbidge C, Hogan DB. Fear of Falling in Older Adults: A Scoping Review of Recent Literature. *Can Geriatr J.* 2021 Dec 1;24(4):379–94.

For more details on the practical implications and this fascinating subject, we certainly would need to address this matter in another dedicated article.

Reviewer: 3

Dr. Anuj Kumar, Apeejay School of Management

Comments to the Author:
Manuscript is acceptable
Response:
Thank you.

VERSION 2 – REVIEW

REVIEWER	Fadda, Marta Università della Svizzera italiana, Institute of Communication and Health
REVIEW RETURNED	03-Feb-2023
GENERAL COMMENTS	The authors have addressed all my comments. The manuscript is now suitable for publication.